# Short-Term Exposure to Bisphenol A Does Not Impact Gonadal Cell Steroidogenesis In Vitro

**DOI:** 10.3390/cells12111537

**Published:** 2023-06-02

**Authors:** Neena Roy, Clara Lazzaretti, Elia Paradiso, Chiara Capponi, Tommaso Ferrari, Francesca Reggianini, Samantha Sperduti, Lara Baschieri, Elisa Mascolo, Carmela Perri, Manuela Varani, Giulia Canu, Tommaso Trenti, Alessia Nicoli, Daria Morini, Francesca Iannotti, Maria Teresa Villani, Elena Vicini, Manuela Simoni, Livio Casarini

**Affiliations:** 1Unit of Endocrinology, Department of Biomedical, Metabolic and Neural Sciences, University of Modena and Reggio Emilia, 41126 Modena, Italy; neena.roy@unimore.it (N.R.); samantha.sperduti@unimore.it (S.S.);; 2Department of Anatomical, Histological, Forensic and Orthopedic Sciences, Sapienza University of Rome, 00161 Rome, Italy; chiara.capponi@uniroma1.it (C.C.);; 3Center for Genomic Research, University of Modena and Reggio Emilia, 42121 Modena, Italy; 4International PhD School in Clinical and Experimental Medicine (CEM), University of Modena and Reggio Emilia, 42121 Modena, Italy; 5Department of Laboratory Medicine and Pathological Anatomy, Azienda USL/Azienda Ospedaliero-Universitaria of Modena, 41126 Modena, Italy; 6Department of Obstetrics and Gynaecology, Fertility Center, ASMN, Azienda Unità Sanitaria Locale-IRCCS di Reggio Emilia, 42123 Reggio Emilia, Italy; 7Unit of Endocrinology, Department of Medical Specialties, Azienda Ospedaliero-Universitaria of Modena, 41125 Modena, Italy

**Keywords:** Bisphenol A (BPA), LH, hCG, steroidogenesis, testis, ovary

## Abstract

Bisphenol A (BPA) is a ubiquitous, synthetic chemical proven to induce reproductive disorders in both men and women. The available studies investigated the effects of BPA on male and female steroidogenesis following long-term exposure to the compound at relatively high environmental concentrations. However, the impact of short-term exposure to BPA on reproduction is poorly studied. We evaluated if 8 and 24 h exposure to 1 nM and 1 µM BPA perturbs luteinizing hormone/choriogonadotropin (LH/hCG)-mediated signalling in two steroidogenic cell models, i.e., the mouse tumour Leydig cell line mLTC1, and human primary granulosa lutein cells (hGLC). Cell signalling studies were performed using a homogeneous time-resolved fluorescence (HTRF) assay and Western blotting, while gene expression analysis was carried out using real-time PCR. Immunostainings and an immunoassay were used for intracellular protein expression and steroidogenesis analyses, respectively. The presence of BPA leads to no significant changes in gonadotropin-induced cAMP accumulation, alongside phosphorylation of downstream molecules, such as ERK1/2, CREB and p38 MAPK, in both the cell models. BPA did not impact *STARD1*, *CYP11A1* and *CYP19A1* gene expression in hGLC, nor *Stard1* and *Cyp17a1* expression in mLTC1 treated with LH/hCG. Additionally, the StAR protein expression was unchanged upon exposure to BPA. Progesterone and oestradiol levels in the culture medium, measured by hGLC, as well as the testosterone and progesterone levels in the culture medium, measured by mLTC1, did not change in the presence of BPA combined with LH/hCG. These data suggest that short-term exposure to environmental concentrations of BPA does not compromise the LH/hCG-induced steroidogenic potential of either human granulosa or mouse Leydig cells.

## 1. Introduction

Bisphenol A (BPA) is an endocrine disruptor affecting various physiological functions, including reproduction, in both men and women [1]. It is extensively used in the manufacture of polycarbonate plastics, epoxy resins, and in polyester-styrene resins, and used in the food-packaging industry in interior liners for food cans, milk containers, and baby bottles, and in dentistry as a sealant [2,3]. BPA can leach into food and water from its packaging, under various conditions such as pH changes, sterilisation, or increased temperature [4,5,6]. This endocrine disruptor has a half-life of less than 6 h after oral administration [7]; it may be accumulated in human tissues and has been detected in blood, urine, fat, mammary tissue, and in the placenta [8,9,10,11,12]. About 4.4–8.8 µM BPA has also been found in the aspirated antral fluid of women undergoing in vitro fertilisation treatment [11]. 

In females, BPA interferes with ovarian follicle development and ovarian steroidogenesis [1,13,14]. BPA is known to mimic, enhance or inhibit estrogenic signals by binding through oestrogen receptors, ERα and ERβ [15]. However, it inhibited follicle growth and induced atresia in cultured mouse antral follicles through a genomic estrogenic independent pathway [16]. Granulosa cells play a crucial role in ovarian follicle growth, steroidogenesis, and oocyte survival and nourishment. Previous studies have reported that steroid hormone synthesis by granulosa cells has been impaired following BPA exposure [16,17,18,19], possibly via the targeting of Cyp11a1 [14]. Evidence suggests that BPA exerts profound effects even in males, damaging spermatogenesis and steroidogenesis, and Leydig cells were extensively used to study the effects of BPA. Previous reports showed that the effect of BPA on Leydig cells varies according to the dose selected. In pubertal Wistar/ST rats, high-dose exposure to BPA decreased cell Leydig number and expression of steroidogenic enzymes [20], and these results were also independently confirmed in Leydig cells isolated from adult males and exposed to a low dose of BPA [21]. Other in vitro studies in human, mouse, and rat foetal testes have reported the association of BPA with decreased testosterone levels [22]. These issues are due, at least in part, to the deleterious effect exerted by BPA on luteinizing hormone (LH)/choriogonadotropin (hCG) receptor (LHCGR) mRNA levels [23], and second messenger activation [24]. LH and hCG bind to their receptor, activating G_α_s protein which in turn triggers the cyclic adenosine monophosphate (cAMP)/protein kinase A (PKA) pathway, leading to the phosphorylation of extracellular-regulated kinase 1/2 (pERK1/2) and cAMP response element-binding protein (CREB), thereby mediating progesterone and testosterone production and gonadal cell proliferation [25]. Thus, both in vitro and in vivo studies have suggested that BPA exposure alters male steroidogenesis through the perturbation of gonadotropin receptor-mediated intracellular signalling. 

Currently available studies are mostly based on the effect of BPA on male and female steroidogenesis following long-term exposure (e.g., for 48 h [26]), to the compound at micromolar concentrations that are higher than the concentrations found in the environment. Very few data are available on the short-term effect of BPA on male and female reproduction in the presence of gonadotropins [24]. Hence, in the current study, we evaluated the impact of short-term exposure to BPA on LH/hCG-mediated signalling in two steroidogenic cell models, i.e., the mouse tumour Leydig cell line mLTC1, and the human primary granulosa lutein cells (hGLC).

## 2. Materials and Methods

### 2.1. Human Samples and Patients’ Selection

Human primary granulosa lutein cells (hGLC) were isolated from the follicular fluid aspirate of about forty women undergoing oocyte retrieval for assisted reproduction at the Santa Maria Nuova Hospital (Reggio Emilia, Italy). Patients matching the following criteria were included in this study: absence of any endocrine abnormalities, severe viral or bacterial infections, and age between 25 and 45 years. The study was approved by the local ethics committee (documents ‘Protocollo n. 2017/0015890 del 26/06/2017′, ‘Protocollo n2018/0080377 del 16/07/2018′, ‘Protocollo n. 2018/0080389 del 16/07/2018′, ‘Protocollo n. 2019/0009846 del 24/01/2019′ and ‘Protocollo n. 2019/0048906 del 23/04/2019′), and written consent was obtained from each patient.

### 2.2. Isolation and Culture of Human Granulosa Lutein Cells 

hGLC was isolated from the follicular fluid, using methods previously described [27,28,29]. Briefly, hGLCs were purified using a 50% Percoll gradient (GE Healthcare, Little Chalfont, UK) to separate them from other cellular components following centrifugation. Haemolysis buffer was added to remove any red blood cell contamination, and this reaction was blocked by adding medium containing DMEM/F12 (Gibco, Thermo Fisher Scientific, Waltham, MA, USA), 10% fetal bovine serum (FBS), 2 mM L-glutamine, 100 IU/mL penicillin, 0.1 mg/mL streptomycin (all from Thermo Fisher Scientific, Waltham, MA, USA) and 250 ng/mL Fungizone (Merck KGaA, Darmstadt, Germany). Blood red cell debris was excluded by centrifugation, and hGLC was washed in Dulbecco’s phosphate-buffered saline (DPBS; Merck KGaA). Finally, the cells were resuspended and cultured in the medium for 6 days to allow them to recover the expression of gonadotropin receptors [30]. After 6 days of culturing, cells were serum-starved overnight and used for analyses.

### 2.3. mLTC1 Cell Line

Mouse Leydig tumour cells (mLTC1) are a commonly used and validated model for studying Leydig cell steroidogenesis [16,31,32,33,34]. mLTC-1 cells (ATCC CRL-2065, LCG Standards, Molsheim, France) were available in-house [34,35] and grown in RPMI 1640 medium supplemented with 10% FBS, 100 IU/mL penicillin, 0.1 mg/mL streptomycin, 2 mM glutamine, and 1 mM HEPES, at 37 °C and 5% CO_2_.

### 2.4. Dose-Finding Experiments for BPA Concentration 

We performed a cell viability assay using 3-(4,5-dimethylthiazol-2-yl)-2,5-diphenyltetrazolium bromide (MTT) to select the best BPA concentrations to be used for the in vitro study. The concentrations used ranged between 1 fM and 1 mM, and we found that 1 mM BPA is toxic to cells (Appendix A). Hence, we chose non-toxic 1 nM and 1 µM BPA concentrations, which are biologically relevant, having been reported present in the environment [36] and even in follicular fluid [11].

### 2.5. cAMP Production

hGLC and mLTC1 cells were seeded in 96-well plates (2 × 10^4^ cells/well). After 24 h, cells were treated for 20 min with the phosphodiesterase inhibitor 3-isobutyl-1-methylxanthine (IBMX) (Sigma-Aldrich, St. Louis, MO, USA). This was followed by 3 h exposure of hGLC to BPA, and 1 h exposure of mLTC1 to BPA, in the presence or absence of the three times (3×) the 50% effective concentration (EC_50_) of LH (1500 pM) and hCG (300 pM) [27,33,37]. The reactions were then stopped by removing the media and rapidly freezing the cells at −80 °C. The next day, samples were collected in 30 µL of the phosphate-buffered saline (PBS), and the total cAMP produced was measured using a cAMP-Gs Dynamic kit for homogeneous time-resolved fluorescence (HTRF), following the manufacturer’s instruction (Cisbio, Codolet, France).

### 2.6. Western Blotting 

hGLC and mLTC1 cells were seeded in 24-well plates (1 × 10^5^ cells/well) and serum-starved 12 h before treatments. The cells were treated with 1 nM and 1 µM BPA, in the presence or absence of 3xEC_50_ LH and hCG. Cells treated for 15 min with phorbol 12-myristate 13-acetate (PMA) served as a positive control. Untreated cells were added as negative controls. The reactions were stopped by removing the media and immediately lysing cells in RIPA Laemmli buffer containing protease and phosphatase inhibitors. Phospho-p38 MAPK, phospho-ERK1/2, phospho-CREB and total ERK1/2 were evaluated via Western blotting using validated specific antibodies and protocols, as previously described [27,37,38]. The signal was developed using a chemiluminescent detection solution (Cyanagen, Bologna, Italy) and acquired using an image analysis system (VersaDoc Imaging System and QuantityOne software 4.6; Bio-Rad Laboratories, Inc., Hercules, CA, USA). Signals were semi-quantified using ImageJ software (National Institutes of Health, Bethesda, MD, USA).

### 2.7. Gene Expression 

5 × 10^4^ cells/well of hGLC and mLTC1 cells were seeded in a 24-well plate and serum-starved overnight. Cells were treated for 8 and 24 h with BPA at 1 nM and 1 µM concentrations, in the presence or in the absence of 3xEC_50_ LH and hCG. Treatments were stopped by removing the media and immediately freezing the cells at –80 °C. The total RNA was extracted using the phenol–chloroform method using RNA Extracol (EURx Sp. z o.o., Gdańsk, Poland). The extracted RNA was reverse-transcribed using Multiscribe reverse transcriptase (Applied Biosystems, Thermo Fisher Scientific, Waltham, MA, USA), and the gene expression was evaluated via real-time PCR. The following genes were evaluated in hGLC: *STARD1* (fw 5′-AAGAGGGCTGGAAGAAGGAG-3′; rev 5′-TCTCCTTGACATTGGGGTTC-3′), *CYP17A1* (fw 5′-AGCCGCACACCAACTATCAG-3′; rev 5′-GCAAACTCACCGATGCTGGA-3′) and *CYP19A1* (fw 5′-TACATTATAACATCACCAGCATCG-3′; rev 5′-TCATAATTCCACACCAAGAGAA-3′). The genes evaluated in mLTC1 were *Stard1* (fw 5′-ACAGACTCTATGAAGAACTT-3′; rev 5′-GACCTTGATCTCCTTGAC-3′) and *Cyp17a1* (fw 5′-CGAACACCGTCTTTCAATGACC-3′; rev 5′-TGGCAAACTCTCCAATGCTG-3′). The real time data were normalized to the endogenous control: *RPS7* (fw 5′-AATCTTTGTTCCCGTTCCTCA-3′; rev 5′-CGAGTTGGCTTAGGCAGAA-3′) for hGLC [39], and *Hprt* (fw 5′-GCGTCGTGATTAGCGATGATG-3′; rev 5′-TCTCGAGCAAGTCTTTCAGTCC-3′) for mLTC1 cells [37], using the 2^−ΔΔCt^ method [40].

### 2.8. Immunofluorescence 

Some 3.5 × 10^4^ cells/well were seeded onto 3-chamber slides, and serum-starved for 12 h before treatment. The cells were treated with 1 µM BPA, in the presence or absence of LH/hCG. For hGLCs, the treatment was carried out in the presence of 1 µM androstendione in all the conditions. After 24 h of treatment, the media were removed and cells were washed in PBS, fixed for 10 min with 4% paraformaldehyde (PFA) (Electron Microscopy Sciences, Hatfield, PA, USA) at 4 °C, washed twice with PBS, and incubated for 10 min with 0.5% TritonX-100 (Sigma-Aldrich, Merck KGaA, Darmstadt, Germany) at room temperature (RT). The reduction of nonspecific background signal was achieved by incubating cells with 1 M glycine (Sigma-Aldrich), and subsequently, with 5% normal donkey serum (Jackson Laboratories Immuno Research, Ely, UK) at room temperature for 30 min. Cells were incubated for 2 h with mouse anti-StAR antibody (1:100, sc-166821, Santa Cruz Biotechnology, Santa Cruz, CA, USA) at room temperature, and then for 1 h with the secondary antibody Alexa Fluor^®^ 488 AffiniPure Donkey Anti-Mouse IgG (H + L) (1:200, #715-545-150 Jackson Immuno Research) at room temperature. Following extensive washes in PBS, cells were counterstained with DAPI (Invitrogen, Thermo Fischer Scientific, Waltham, MA, USA) and mounted with Vectashield mounting medium (Vector Laboratories, Newark, CA, USA). Digital pictures were acquired using a Zeiss Airyscan 2 confocal microscope. 

### 2.9. Analysis of Steroidogenesis

hGLC and mLTC1 were seeded in 48-well plates at a concentration of 3 × 10^4^ cells/well, and serum-starved 12 h before treatment. Cells were treated with BPA in the presence or absence of LH and hCG, for 8 and 24 h. For hGLCs, the treatment was carried out in the presence of 1 µM androstenedione as a substrate for oestrogen production [41]. The media were collected and analyzed for progesterone and oestradiol concentrations in hGLCs, and for progesterone and testosterone in mLTC1. The analysis was carried out using an Architect Immunoassay Analyzer (Abbott Laboratories, Abbott Park, IL, USA).

### 2.10. Statistical Analysis

Statistic analyses were performed using GraphPad Prism 9 (GraPhPad Software Inc., San Diego, CA, USA). The results were analysed using the Kruskal–Wallis test, after testing for normality with D’Agostino and Pearson tests. Values of *p* < 0.05 were considered significant.

## 3. Results

### 3.1. Cell Signalling Analysis

In hGLC and mLTC1 cells, we evaluated the effect of BPA on cAMP production mediated by LH and hCG. While the treatment with gonadotropins induced an intracellular cAMP increase (Figure 1; Kruskal–Wallis test, *p* < 0.05), no different effects between gonadotropin administered alone or in combination with 1 nM and 1 µM BPA were found (*p* ≥ 0.05). Similar results were obtained in a control experiment (*p* ≥ 0.05), where hGLC and mLTC1 cells were treated with the adenylyl cyclase activator forskolin, or with the Gαs protein activator cholera toxin, in the presence of BPA (Appendix A). These data suggest that early LH/hCG-induced effectors could not be hindered by BPA in granulosa and Leydig cells. 

The effect of BPA on the gonadotropin-induced phosphorylation of molecules downstream of cAMP, such as ERK1/2, CREB and p38MAPK, was evaluated in both hGLC and mLTC1 cells. PMA-treated cells served as a positive control. Western blot analysis revealed that LH/hCG alone and together with BPA induced similar phosphorylation patterns in ERK1/2, CREB and p38 MAPK in both the models in vitro (Figure 2; Kruskal–Wallis test, *p* ≥ 0.05). Indeed, both the hormones, as well as PMA, induced phospho-protein activation. However, the presence of 1 nM and 1 µM BPA did not impact the results both in hGLC and mLTC1 cells.

### 3.2. Gene Expression Analysis

The effects of BPA on the 8 and 24 h expression of genes regulating LH/hCG-induced steroid synthesis were evaluated in hGLC, using real-time PCR. Gonadotropins upregulated the 8 h expression of *STARD1* and *CYP19A1* target genes (Figure 3; Kruskal–Wallis test, *p* < 0.05), while *CYP17A1* gene expression increased after 24 h (*p* < 0.05). However, gene expression patterns were not altered in hGLC treated with LH/hCG alone, and in combination with 1 nM and 1 µM BPA (*p* ≥ 0.05).

Gene expression analysis was performed in mLTC1, where the effects of BPA addition to cell treatment with gonadotropins were investigated. *Stard1* and *Cyp17a1* gene expression was evaluated using real-time PCR; this was not the case for the *Cyp19a1* gene, which is silent in the Leydig cell [42]. In mLTC1 cells, LH/hCG significantly increased both 8 and 24 h *Stard1* and *Cyp17a1* gene expression (Figure 4; Kruskal–Wallis test, *p* < 0.05). However, gonadotropin–BPA cotreatment is not linked to any change in gene expression patterns at both timepoints evaluated (*p* ≥ 0.05).

We also investigated the effect of BPA on StAR protein expression, as a key regulator of steroidogenesis. hGLC and mLTC1 cells were treated for 24 h by LH/hCG, in the presence or in the absence of BPA, and the StAR protein was evaluated using immunostaining. As expected, gonadotropins treatments increased the cytoplasmic levels of StAR immunoreactivity (Figure 5A–C). However, in line with our gene expression data, no changes were induced by BPA co-treatments (Figure 5D–F,J,L). 

### 3.3. BPA Did Not Alter Steroidogenesis

To further assess the effect of BPA on steroidogenesis, we also measured the production of progesterone and oestradiol in the culture media using hGLC, and progesterone and testosterone in the culture media using mLTC1. To this end, cells were treated for 8 and 24 h with gonadotropins, in the presence or in the absence of BPA. LH and hCG enhanced progesterone production via hGLC at both the timepoints evaluated, while oestradiol synthesis by hGLC required 24 h (Figure 6; Kruskal–Wallis test, *p* < 0.05, n = 5). However, BPA did not significantly impact steroid production (*p* ≥ 0.05). These results match those obtained from the culture media of mLTC1 cells, wherein the production of progesterone and testosterone by mLTC1 was increased upon cell treatment with LH and hCG (Figure 7; Kruskal–Wallis test, *p* < 0.05), and the addition of BPA did not change steroid levels (*p* ≥ 0.05).

## 4. Discussion

We found that short-term exposure to BPA at concentrations comparable to those found in human biological fluids [11] did not produce any effect on the LH/hCG-mediated steroidogenic pathway, both in hGLC and mLTC1 cells. In this study, we tested two environmental BPA concentrations, i.e., 1 nM and 1 µM, and performed cell signalling, gene expression, and steroidogenesis analyses. All the endpoints evaluated herein are known targets of gonadotropin action, and were not affected by the presence of BPA. These results suggest that short-term exposure to BPA does not impact the steroidogenic activity of gonadal cells, in vitro.

BPA is a ubiquitous environmental contaminant in Europe, America, Asia, and Australia, with concentrations ranging from 10 to >100,000 μg/kg dry weight [36]. The compound may induce adverse reproductive disorders in both men and women, although the data available do not converge in a single direction [26]. It is a common view that in females, BPA interferes with ovarian follicle development and steroidogenesis [13,43]. In cultured human luteinized granulosa cells, relatively high BPA concentrations (2 and 20 µg/mL, corresponding to about 8.7 and 87.6 µM) decreased oestradiol and progesterone production within 48 h [44]. The molecular mechanism by which BPA can impair steroidogenesis should involve both the mRNA and protein expression of steroidogenic enzymes, such as cholesterol monooxygenase (CYP11A1), 3β-hydroxysteroid dehydrogenase (3β-HSD), and aromatase, without affecting StAR expression [44]. However, the same study found opposite results under lower concentrations. Acute exposure to a similar concentration range of the compound (about 0.9–90.0 µM) did not alter the 48 h steroidogenesis in granulosa cells in vitro, suggesting that the main impact of BPA on ovarian cells could be exerted under conditions of chronic exposure [44]. We observed a similar effect in LH/hCG-induced progesterone and oestradiol levels when hGLC cells were treated with 1 nM and 1 µM BPA, which did not produce any effect on steroidogenesis and confirmed previous observations [44]. Our study indicates that the endocrine disruptor did not change cAMP levels, phospho-protein activation, and the expression of gonadotropin target genes and steroid synthesis. Overall, our results disagree with previous in vitro studies demonstrating the disruptive effects of BPA in human granulosa cell steroidogenesis [45,46]. For instance, higher BPA concentrations (40–100 µM) than those used in the present study induced the upregulation of aromatase expression and oestrogen synthesis [45]. Another study testing the 48 h exposure of cumulus granulosa cells to BPA revealed that the compound increases progesterone and decreases oestradiol synthesis [45], while experiments in a human ovarian granulosa cell-derived cell line, KGN, demonstrated that fM–pM doses of BPA increase oestradiol production. Taken together, data provided by in vitro studies are controversial, and do not allow us to achieve conclusions about the short-term effect of BPA on steroidogenesis. We may hypothesize that several perturbing factors, such as genetic background [47], polyphenol-containing plastic materials used for cell culture [48], or the additive effect of reagents for in vitro studies [49], could modulate or even mask the eventual perturbation of the endocrine signal exerted by BPA.

In males, BPA has been shown to disrupt steroidogenesis [50]. In vitro studies using mouse, rat and human cultured foetal testes have demonstrated decreased testosterone production induced by treatment with 1 fM to 10 µM BPA [22]. Interestingly, BPA affected only the functions of humans, but not rodent Leydig cells, suggesting that the endocrine disruption exerted by this compound is not necessarily similar among mammals. Moreover, the action of BPA is differently exerted according to the given experimental conditions. In cultured mLTC1 cells, exposure to pM–µM BPA concentrations inhibited hCG-induced cAMP and progesterone production [24]. These results do not align with those described in the present study, wherein we did not find any significant effect on LH/hCG-stimulated cAMP and progesterone production to be linked to the presence of 1 nM and 1µM BPA. The reason for this difference may be related to the exposure time and the concentrations that we used. The authors exposed cells to higher concentrations of BPA for 48 h [24], while we used BPA at lower concentrations, with exposure for 1 h for cAMP and for both 8 and 24 h exposure for progesterone production. Rather, our results match those of previous studies reporting that exposure to 1 nM–1 µM BPA for 4 or 24 h did not impact testosterone levels, regardless of the presence of hCG [51], and that even 10 µM BPA failed to inhibit hCG-induced upregulaton of the *Stard1* gene [52]. Again, these findings point out the relevance of careful experimental settings for investigating the effects of endocrine disruptors in vitro.

## 5. Conclusions

Steroidogenic signalling plays a critical role in fertility. BPA, an endocrine-disrupting compound, is known to affect steroidogenic pathways affecting male and female fertility. In this study, we observed that short-term exposure to environmental concentrations of BPA does not compromise the steroidogenic potential of either human granulosa or mouse Leydig cells. Our findings add to those that report weak or no acute action of BPA in gonadal cells. However, several papers reporting opposite results suggest that the issue must be further investigated, using an experimental setting aiming to exclude potential confounding factors that may affect results.

## Figures and Tables

**Figure 1 cells-12-01537-f001:**
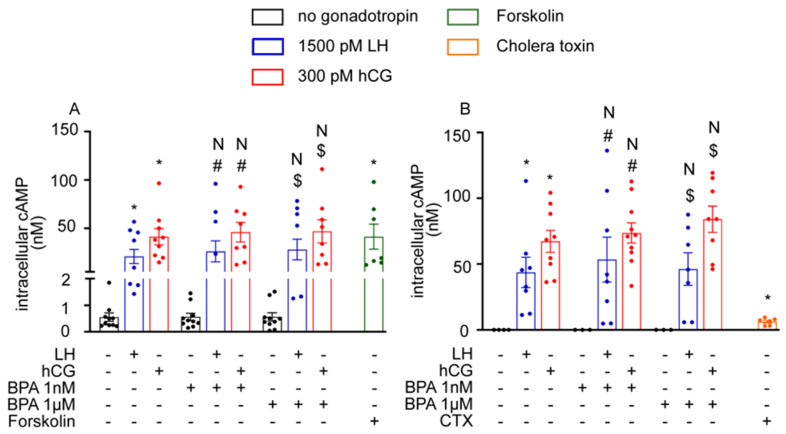
Effect of BPA on cAMP production in vitro: (**A**) in human granulosa cells (hGLC); (**B**) in mouse Leydig cells (mLTC1). cAMP was measured using an HTRF assay. Forskolin and cholera toxin served as the positive control in hGLC and mLTC1 cells, respectively. Data are represented as mean ± SEM. CTX = cholera toxin. *, Significantly different vs. “no gonadotropin” alone; # vs. “no gonadotropin” + BPA 1 nM; $ vs. “no gonadotropin” + BPA 1 µM; N, not significantly different vs. LH/hCG treatment in the absence of BPA (Kruskal–Wallis test, *p* ≥ 0.05).

**Figure 2 cells-12-01537-f002:**
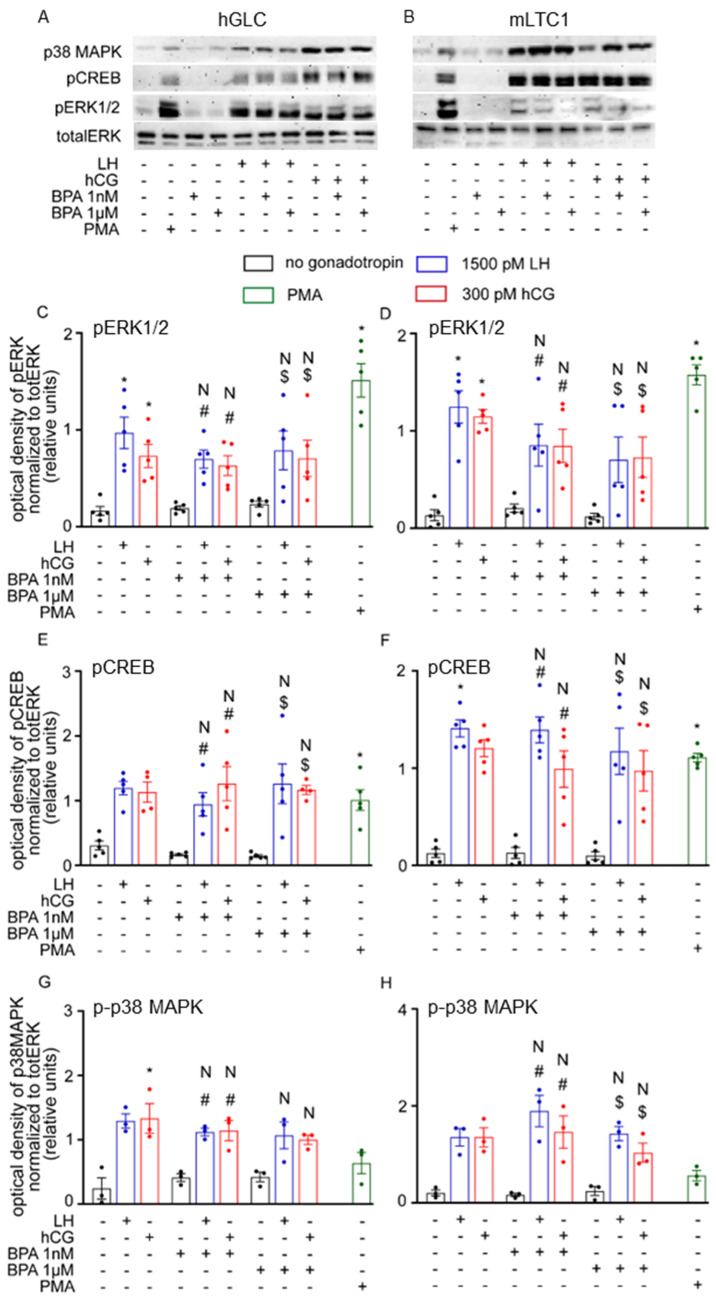
Effect of BPA on activation of pERK, pCREB and p38MAPK: Western blotting analyses in hGLC (**A**) and in mLTC1 (**B**) after exposure to BPA without or with hCG or LH. Relative semi-quantification of the activation of pERK (**C**,**D**), pCREB (**E**,**F**) and p38MAPK (**G**,**H**) in hGLC and mLTC1 cells, respectively. Data are represented as mean ± SEM. *, Significantly different vs. “no gonadotropin” alone; # vs. “no gonadotropin” + BPA 1 nM; $ vs. “no gonadotropin” + BPA 1 µM; N, not significantly different vs. LH/hCG treatment in the absence of BPA (Kruskal–Wallis test, *p* ≥ 0.05).

**Figure 3 cells-12-01537-f003:**
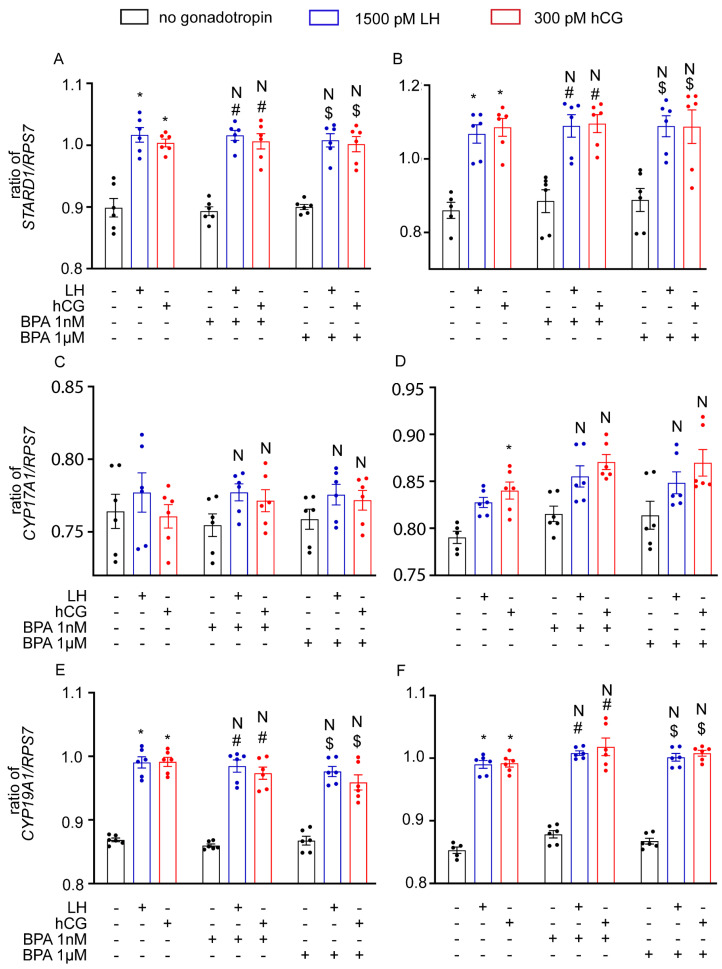
Effect of BPA on steroidogenic gene expression in hGLC after 8 h (**A**,**C**,**E**) and 24 h (**B**,**D**,**F**) of treatment. Genes: *STARD1* (**A**,**B**), *CYP17A1* (**C**,**D**) and *CYP19A1* (**E**,**F**). Data are represented as mean ± SEM. *, Significantly different vs. “no gonadotropin” alone; # vs. “no gonadotropin” + BPA 1 nM; $ vs. “no gonadotropin” + BPA 1 µM; N, not significantly different vs. LH/hCG treatment in the absence of BPA (Kruskal–Wallis test, *p* ≥ 0.05).

**Figure 4 cells-12-01537-f004:**
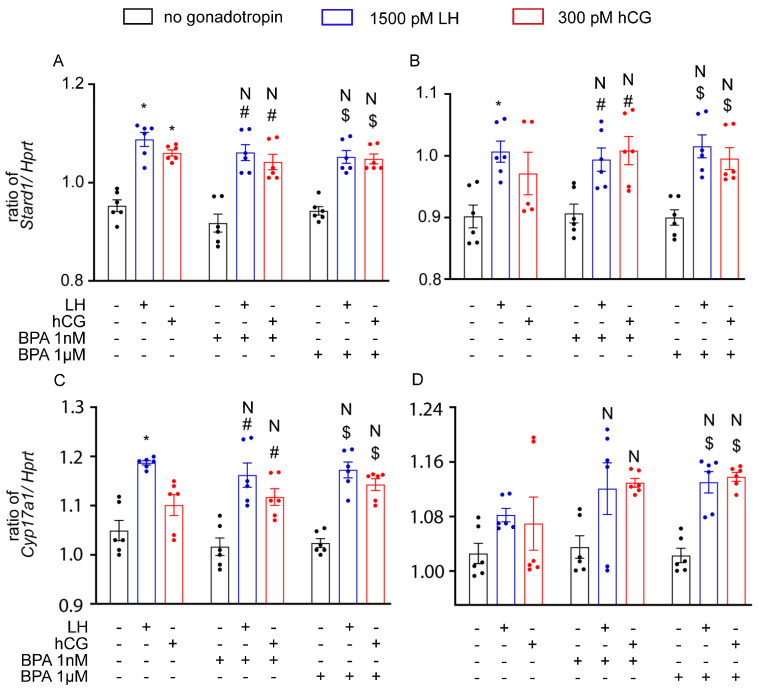
Effect of BPA on steroidogenic gene expression in mLTC1 after 8 h (**A**,**C**) and 24 h (**B**,**D**). Genes: *Stard1* (**A**,**C**) and *Cyp17a1* (**B**,**D**). Data are represented as mean ± SEM. *, Significantly different vs. “no gonadotropin” alone; # vs. “no gonadotropin” + BPA 1 nM; $ vs. “no gonadotropin” + BPA 1 µM; N, not significantly different vs. LH/hCG treatment in the absence of BPA (Kruskal–Wallis test, *p* ≥ 0.05).

**Figure 5 cells-12-01537-f005:**
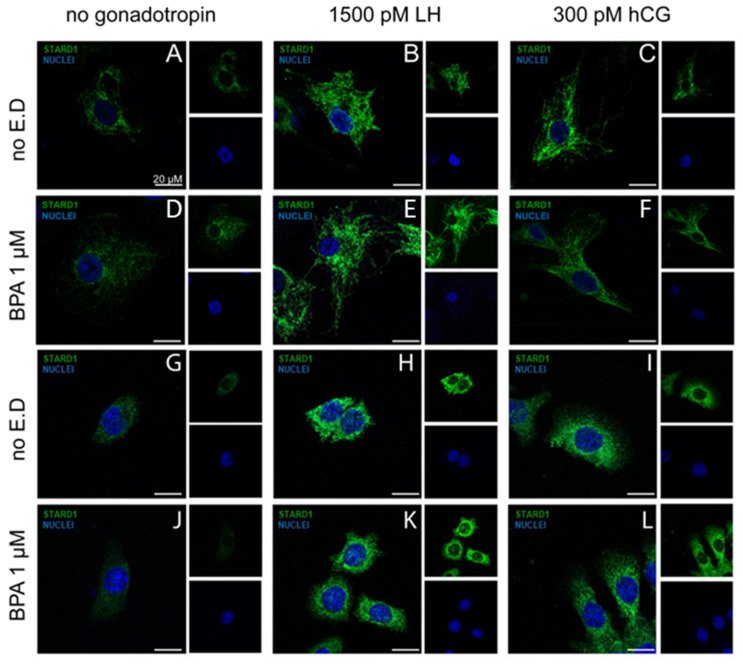
Immunofluorescence images of the effect of BPA on StAR protein expression in LH/hCG-treated hGLC (**A**–**F**) and in mLTC1 cells (**G**–**L**). Cells were treated 24 h with 1 µM of BPA, in the absence or presence of LH/hCG. Images are representative of three independent experiments (magnification 40×. Subfigure size is reduced to 50%).

**Figure 6 cells-12-01537-f006:**
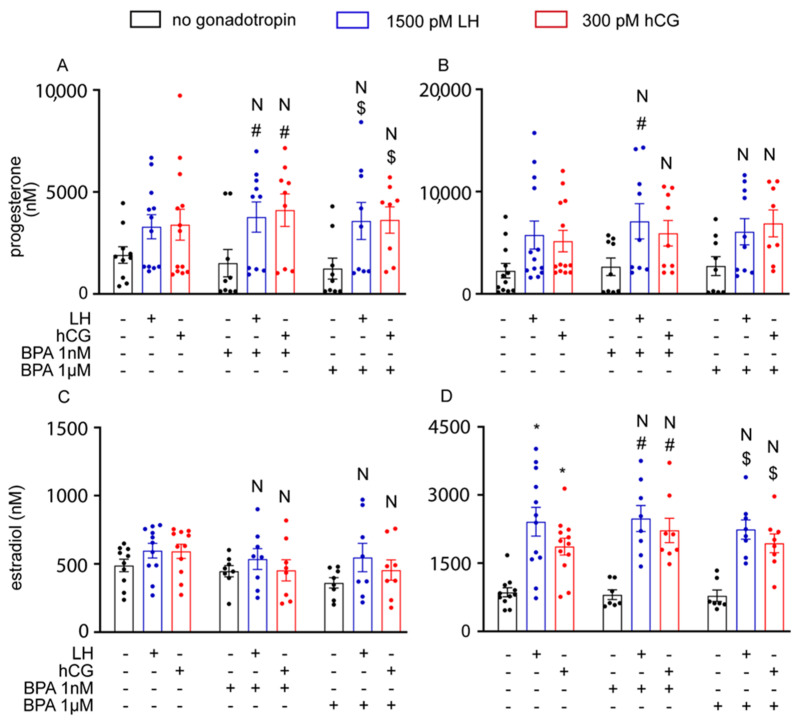
The effect of BPA on progesterone and oestradiol levels in hGLC cells after 8 h (**A**,**C**) and 24 h (**B**,**D**). Data are represented as mean ± SEM. *, Significantly different vs. “no gonadotropin” alone; # vs. “no gonadotropin” + BPA 1 nM; $ vs. “no gonadotropin” + BPA 1 µM; N, not significantly different vs. LH/hCG treatment in the absence of BPA (Kruskal–Wallis test, *p* ≥ 0.05).

**Figure 7 cells-12-01537-f007:**
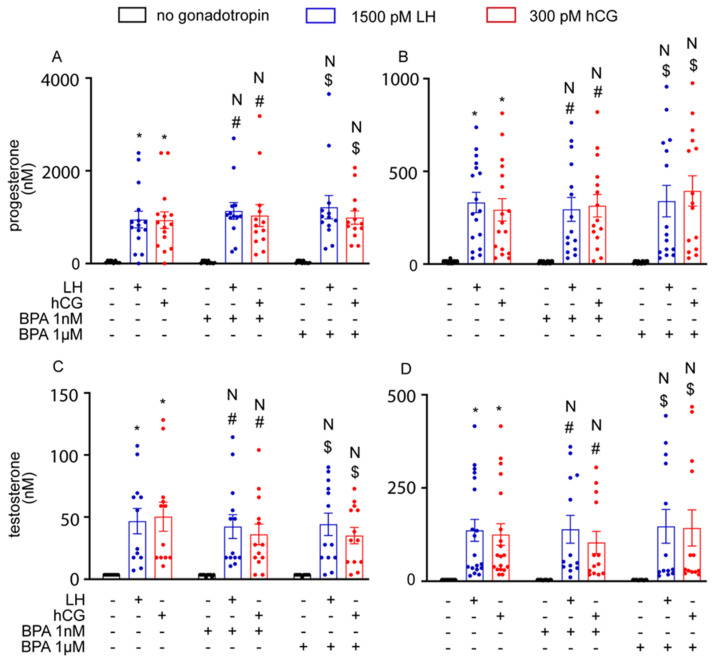
The effect of BPA on progesterone and testosterone levels in mLTC1 cells after 8 h (**A**,**C**) and 24 h (**B**,**D**). Mean ± SEM. *, Significantly different vs. “no gonadotropin” alone; # vs. “no gonadotropin” + BPA 1 nM; $ vs. “no gonadotropin” + BPA 1 µM; N, not significantly different vs. LH/hCG treatment in the absence of BPA (Kruskal–Wallis test, *p* ≥ 0.05).

## Data Availability

All other data are available from the corresponding author upon reasonable request.

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
