# Peer review of "Short-Term Exposure to Bisphenol A Does Not Impact Gonadal Cell Steroidogenesis In Vitro"

_cells, 2023, doi:10.3390/cells12111537_

Round 1
Reviewer 1 Report
The group of L. Casarini attempts to investigate whether short-term (8 and 24h) exposure of ‘male and female’ steroidogenic cell models to BPA affects LH/hCG-mediated signaling. To assess this, the authors performed series of in vitro experiments using human primary granulosa lutein cells (hGLC) and mouse tumor Leydig cell line (mLTC1). They found that “short-term exposure to low concentrations of BPA does not compromise the steroidogenic potential of either human granulosa or mouse Leydig cells.” Overall, the methodology is adequate to accomplish research goals, however, experimental design and statistical analyses raise serious concern. I am afraid that the results are not very convincing, sometimes contradicting the available literature and far from conclusive; no concrete conclusions could fairly be drawn from these studies about the physiological role of BPA in reproductive function.
Specific Comments
· I have some serious concern about the statistical analysis. Why the authors decided to use the Kruskal-Wallis test? This is a nonparametric test dedicated to compare three or more unpaired or unmatched groups, while in these in vitro exps groups are related. I agree that if the data are not normally distributed, a non-parametric test should be used, but in this case, repeated measures ANOVA (more robust statistics) followed by e.g. Dunn’s post-hoc test is most appropriate. Continuing the above, please explain why there is unequal number of cultures (“n”) for different treatments in the same experiment and how it is possible that “n” for controls is smaller than for treatments, for instance in Fig. 2D, n=3 for non-gonadotropin group and n=7 for 1 uM BPA that is not acceptable. I assume that each “n” represents hGLC obtained from individual cell isolation, while mLTC-1 cells from different passages, so in my opinion the same “n” should be used to make any comparisons/analyses between groups.
· It is unclear how many women were involved in the study, and how the cells were distributed for specific examinations. Same for mLTC1. These should be clearly stated.
· Could you confirm that mLTC1 cells are a reliable model for study steroidogenesis in Leydig cells? The state of healthy and cancer cells are quite different; therefore, I would be careful in interpreting the data obtained on mLTC1 cells.
· Did the authors measure the levels of urinary or blood or follicular fluid BPA in donors? As BPA reduces E2 response to gonadotropin stimulation during IVF, it would be interesting to see whether there is any correlation between, e.g. BPA and the number of oocytes retrieved and E2 levels. I also wonder if all primary cells from different women exhibit similar results.
· How exactly did you calculate/normalize the qPCR data, in other words what algorithm/test was used? I have never seen gene expression expressed as the ratio of a reference gene to a target gene. Please explain it.
· Fig. 5 and the legend need some clarification and improvement: What is “E.D”? Specify magnification. Negative controls for immunofluorescence should be provided. In the merged image in Fig. 5E I don’t see nuclear staining. In Fig. 5J is hard to see anything.
· In figure legends define all symbols, e.g. *, #, $, N etc. that you used in the figures. Also provide “n” for each experiment.
· Specify the term “long-term exposure”.
· L36-37: If I understand correctly, P4, E2 and T were measured in culture medium, and not in the cells; please correct this information here and in the text.
· Why the authors state that 1 uM BPA is a low concentration? What is the so-called physiological concentration of BPA in follicular fluid? What is a half-life of BPA in biological fluids?
· Please check all refs carefully, e.g. in L55 you describe human studies but cite a paper [ref.13] that is dedicated to animals.
· L54: How does BPA actually affect the ovarian follicle? There is some evidence in the literature that BPA is an endogenous estrogen mimic, capable of binding to estrogen receptors, eventually leading to disrupted ovarian response. I think it would be reasonable do add/discuss this data.
· L77: The meaning of the following part of the text “environmentally relevant concentrations” is ambiguous. A more specific sentence would be more useful.
· L83-85: remove
· L87: change into “intracellular cAMP”
· L79: I think the goal of this study needs some improvement. It would be better to emphasize that “we evaluated the impact of short-term BPA…”
· L102: change as follows “production in vitro in: A) human…”
· In Figure 2, it would be good to place the names of the analyzed proteins on the Y axis
· In Suppl Fig. 2A there is no bar for control.
Author Response
The response file is attached, thank you.

Reviewer 2 Report
This manuscript explored the effect of low concentration, acute BPA exposure on steroidogenesis using two different kinds of gonadal cells. This is meaningful. As the authors also set positive control, the current negative results are trustful. However, I found this manuscript need to be improved, especially in the result section.
1. In "Result" part, I think line 83-85 has no relation with the following results.
2. For 2.1. cell signaling analysis, I didn't find any result description about Cholera toxin. Besides, the description of WB result is too simple, which is not clear enough for understanding.
3. There are many obvious errors in Figure1, 1). The blue color of columns in the left bar graph is incorrect. 2). There is no Cholera toxin group right below the bar graphs. 3). I can't figure out what "*, #, N, $" mean, not only in Figure 1. 4). Also need to make sure the position of the "*, #, N, $".
4. For Figure 2 A and B, 1). The images are not clear enough. 2). I found you didn't use the commonly used housekeeping protein, and instead you used totalERK. I think it's ok to normalize pERK1/2 to totalERK. But for other proteins, I don't think totalERK can be used as housekeeping protein. 3). The lower band of totalERK in Fig. 2B gradually disappear. 4). Y axis should be named as the name of the protein to make the manuscript easier to read. 5). I also noted in more than one figure, there is only 2 samples in some group. Please increase the sample size.
5. For mRNA part, why do you choose PRS7 and Hprt as endogenous control? Besides, what do you mean the ration of PRS7/STARD1? How did you calculate it?
6. For Line 167-169, how did know "10 to100000 ug/kg dry weight" is equal to "44 nM-440 nM"?
7. Before treatment, have you ever checked the cell density, as you seeded different numbers of cells in different plates?
8. For Line 61-62, what "cell Leydig number" means?
9. In Supplementary figure 1, the legend should be filled with corresponding colors.
good
Author Response
Reponse to the Reviewer 2 is attached, thank you.

Round 2
Reviewer 1 Report
I thank the authors for addressing my comments. The updated manuscript is improved and does not require further corrections in my opinion.
Author Response
I thank the authors for addressing my comments. The updated manuscript is improved and does not require further corrections in my opinion.
We thank the Reviewer to have appreciated our study.
Reviewer 2 Report
I can find obvious changes in the manuscript, which makes it much easier to read. I still found some issues need to be improved.
1. In Fig.2, the WB part is clear, but Fig.2 C-H are not clear enough.
2. Still WB part, I think "CREB" is necessary, then you can calculate the ratio of "pCREB / CREB", rather than "pCREB / totERK". As you mentioned, you checked p-p38 MAPK, but I didn't find the results of p-p38, you also need to calculate the ratio of "p-p38 MAPK / p38 MAPK", rather than "p-p38 MAPK / totERK".
3. In Fig.4A and C, the sample size of BPA 1uM is still 2, you need to add more samples.
Author Response
I can find obvious changes in the manuscript, which makes it much easier to read. I still found some issues need to be improved.
We are grateful to this Reviewer for having provided other suggestions to improve the manuscript.
- In Fig.2, the WB part is clear, but Fig.2 C-H are not clear enough.
We have modified Fig 2. C-H indicating names of phospho-proteins and cell lines in all panels. It should be clearer now.
- Still WB part, I think "CREB" is necessary, then you can calculate the ratio of "pCREB / CREB", rather than "pCREB / totERK". As you mentioned, you checked p-p38 MAPK, but I didn't find the results of p-p38, you also need to calculate the ratio of "p-p38 MAPK / p38 MAPK", rather than "p-p38 MAPK / totERK".
We agree with the reviewer that phospho-proteins could be elegantly normalized over the corresponding total protein. However, total ERK is an established, reliable and largely used normalizer for all those phospho-proteins in our cell models, as indicated by previous validations referenced in the “Materials and Methods” section (line 144). It is based on the fact that total ERK expression levels are overall similar to those of CREB, p38 MAPK, etc, leading to the correspondence between total ERK and phospho-protein concentration variations. It is the same principle of using beta-Actin as normalizer for several other proteins, performed in light of similar expression levels. Instead, the same relationship is missing between molecules having widely different expression levels than phospho-proteins, such as beta-Actin, Tubulin or GAPDH. The latters are indeed relatively highly expressed and are not acceptable phospho-protein normalizers (with some cell type-dependent exceptions: Chen et al. PLoS One. 2012;7(8):e42598). In any case, the reliability of our signalling pathway analysis by Western blotting is further demonstrated by results that are in line with cAMP, gene expression and steroidogenesis data, which univocally showed the lack of short-term BPA effect.
- In Fig.4A and C, the sample size of BPA 1uM is still 2, you need to add more samples.
We have increased the sample size, thank you.